# Prognostic Utility of CD47 in Cancer of the Uterine Cervix and the Sensitivity of Immunohistochemical Scores

**DOI:** 10.3390/diagnostics13010052

**Published:** 2022-12-24

**Authors:** Angel Yordanov, Velizar Shivarov, Stoyan Kostov, Yonka Ivanova, Polina Dimitrova, Savelina Popovska, Eva Tsoneva, Mariela Vasileva-Slaveva

**Affiliations:** 1Department of Gynaecological Oncology, Medical University Pleven, 5800 Pleven, Bulgaria; 2Research Institute, Medical University Pleven, 5800 Pleven, Bulgaria; 3Department of Gynecology, St. Anna University Hospital, Medical University—Varna “Prof. Dr. Paraskev Stoyanov”, 9000 Varna, Bulgaria; 4Department of Pathology, Medical University–Pleven, 5800 Pleven, Bulgaria; 5Department of Obstetrics and Gynecology, Shterev Hospital, 1000 Sofia, Bulgaria; 6Department of Breast Surgery, Shterev Hospital, 1000 Sofia, Bulgaria

**Keywords:** cancer of the uterine cervix, CD47, expression, immunohistochemical scores

## Abstract

Introduction: Cancer of the uterine cervix (CUC) is still one of the most frequent oncological diagnoses in women. The specific interactions between the tumor cells of CUC and the cells and tissues in the tumor microenvironment can affect cancer cells’ invasive and metastatic potential and can modulate tumor’s progression and death. CD47 is a trans-membranous immunoglobulin, expressed in many cells. It protects the cells from being destroyed by the circulating macrophages. Aim: We aimed to evaluate the prognostic role of CD47 expressed in the tumor tissues of patients with CUC for tumor progression and to find the most sensitive immunohistochemical score for defining the cut-off significantly associated with tumor biology and progression. Materials and methods: Paraffin-embedded tumor tissues from 86 patients with CUC were included in the study. Clinico-morphological data for patients, such as age and stage at diagnosis according to FIGO and TNM classification, were obtained from the hospital electronic medical records. Immunohistochemical staining was performed with rabbit recombinant monoclonal CD47 antibody (Clone SP279). The final result was interpreted based on three reporting models in immunohistochemistry: H-score, Allred score and combined score. Results: The expression of CD47 was higher in tumors limited in the cervix compared with those invading other structures, and it did not depend on the nodal status. The results of immunohistochemical staining were similar regardless of which immunohistochemical method was used. The most significant correlation with TNM stage was observed with the H-score (*p* = 0.00018). The association with the Allred and combined score was less significant, with p values of 0.0013 and 0.0002, respectively. Conclusion: The expression of CD47 in the cancer cells is prognostic for tumor invasion in the surrounding structures, independent of lymph node engagement. The H-score is the most sensitive immunohistochemical score to describe tumor stage. To the best of our knowledge, this is the first study evaluating the significance of CD47 expression in CUC.

## 1. Introduction

In many countries, cancer of the uterine cervix (CUC) is still one of the most frequent oncological diagnoses in women. Globally, approximately 600,000 new cases are diagnosed and over 300,000 deaths are observed every year from this disease [1]. CUC is significantly more often diagnosed in developing countries, the second highest cancer-related mortality is recorded [2].

It has been proven that tumor cells have specific interactions with the tissues in the tumor microenvironment. These interactions can affect their invasive and metastatic potential, which are the reasons for tumor progression and death [3]. However, the exact interactions between tumor and the healthy cells in the tumor-associated microenvironment are unknown. If those interactions are better understood, this can elucidate the process of CUC progression and indicate new possible prognostic and predictive biomarkers.

Cluster of Differentiation 47 (CD47) is a trans-membranous immunoglobulin, coded by the CD47 *gene* [4]. It is also called integrin-associated protein (IAP) and can be found on the surface of many different types of cells in the body. Its role is to protect the cells from being destroyed by circulating macrophages. CD47 protein combines in a strong signaling complex with another signaling and regulatory protein, SIRPα, also called a “don’t eat me” signal [5]. When the expression of CD47 is decreased in old and sick cells, those cells are attacked by macrophages. It has been proven that CD47 is strongly expressed in different types of tumors, and this has been associated with poor prognoses. Therefore, it has been hypothesized that interventions over CD47 protein can have a therapeutic potential is some diseases [4].

Immunohistochemically, CD47 has been investigated in bone marrow samples of patients with acute myeloid leukemia, melanoma, and ovarian cancer, but to the best of our knowledge, this is the first study of the role of CD47 in CUC [6].

We investigated the expression of CD47 in tumor tissues of patients with CUC in different stages and analyzed the sensitivity of different immunohistochemical scores to define the cut-off most significantly associated with tumor biology and progression.

## 2. Materials and Methods

Paraffin-embedded tumor tissues from 86 patients with CUC, diagnosed between 2015 and 2020 in the Department of Pathology, Medical University Pleven, were included in the study. All patients signed informed consent for their samples to be used for further research. Ethical committee permission was obtained to investigate the role of CD47 in cancer progression (number 656/29.06.2021). Clinico-morphological data for patients were obtained from the electronic database of the department.

### 2.1. Patients Characteristics

We collected data for patient’s age and tumor stage at diagnosis according to FIGO and TNM classifications. The 8th edition of the TNM classification and FIGO classification 2009 was used (Table 1).

Patients were classified according to the FIGO Stage. In FIGO III stage, there were only patients with lymph node metastases: FIGO IIIC. According to the current guidelines for the treatment of CUC for patients in stage FIGO IIIA and FIGO IIIB, primary surgery is not recommended.

### 2.2. Immunohistochemical Scores

Currently, there is no accepted scoring system for CD47. We decided to investigate and compare 3 established methods in immunohistochemistry: H score, Allred score, and the combination of both.

For each patient, we selected one slide with hematoxylin–eosin staining. From the corresponding formalin-fixed and paraffin-embedded tumor block, we stained one section with a 3 µm thickness of CD47 (Clone SP279, Rb, dilution 1:100, Abcam, UK). We used immunohistochemistry with a visualization EnVision ™ FLEX, High pH (DAKO) system and AutostainerLink 48 technique (DAKO). We performed heat-mediated antigen retrieval with citrate buffer, pH 6, before commencing with the IHC staining protocol. As a positive external control, we used a prostate adenocarcinoma tissue included in each run.

The entire tissue section was evaluated at low magnification, then at high magnification, considering two indicators: (I) degree of intensity—absent, weak, moderate, and strong (Figure 1 and Figure 2); and (II) the percentage of positive viable tumor cells. The localization of expression in tumor cells (cytoplasmic/membrane/nuclear) was reported. When determining positivity, only membrane staining was included.

The final result was interpreted based on three reporting models: H-score, Allred score, and combined score. The three systems classify carcinomas into similar, but not identical, groups.

### 2.3. H-Score

For H-score assessment, the following formula was applied:

CD47 H-score = (% of cells stained at weak intensity × 1) + (% of cells stained at moderate intensity × 2) + (% of cells stained at strong intensity × 3).

The resulting scores ranged from 0 to 300, where 300 was equal to 100% of tumor cells stained strongly (3+).

The expression level was categorized according to the median value of the H-score: low (with H-score ≤ 74) or high (with H-score > 74). If there were <1% positive cells with H-score = 0, it considered to be a negative result.

### 2.4. Allred Score

For the Allred score, the following formula was used (Table 2):

Proportion score (PS): 0 (no cells staining); 1 (<1% cells staining); 2 (1–10% cells staining); 3 (11–33% cells staining); 4 (34–66% cells staining); 5 (67–100% cells staining).

Intensity score (IS): 0 (no staining); 1 (weak staining); 2 (moderate staining); 3 (strong staining).

### 2.5. Combined Score

For the combined score we reported two indicators:
Degree of intensity: missing (0 pts), weak (1 pts), moderate (2 pts), strong (3 pts).Percentage of positive tumor cells: no positive cells (0pts), 1–5% (1 pts), 6–25% (2 pts), 26–50% (3 pts), 51–75% (4 pts), 76–100% (5 pts).

The final result was obtained based on the summation of the points from the two categories: negative result, with complete/nearly complete lack of expression (0–2 pts); weak expression (3–6 points); overexpression (7–8 points) (Table 3).

### 2.6. Statistical Methods

The distribution of patients per group was summarized using standard descriptive measures such as counts and percentages (Table 1). Comparisons for CD47 H-score between more than two groups were performed using non-parametric Kruskal–Wallis tests. Two-group comparisons for CD47 H-score were performed using two-sided Wilcoxon rank-sum tests. *p*-values below 0.05 were considered statistically significant. All tests were implemented using the R statistical environment for Windows (version 4.2.0). All plots were generated using R packages *ggpubr* (v. 0.4.0) and *ggplot2* (v. 3.3.5).

## 3. Results

We evaluated the expression of CD47 in patients with CC in different tumor stages in different ways (Table 3).

We analyzed the relationship between CD47 expression and T stage, FIGO stage, and N status, as well as combined CD47 expression levels in patients with different tumor sizes but the same N status. The results from all three different techniques of reporting CD47 expression were similar, and there was no significant difference in the distribution of values in the radical groups according to FIGO stage.

When we use H-score (Figure 3) we get the following results:

When comparing the expression levels in the different T stages, we had a statistically significant difference depending on the infiltration of the tumor in neighboring structures—pT1 vs. pT2 (*p* = 0.00018).

There were no statistically significant differences in CD47 expression depending on the type of adjacent organ infiltrated (vagina or parametrial area) pT2A vs. pT2B (*p* = 0.78) (Figure 3A).

The results were similar when comparing expression levels depending on the FIGO stage (Figure 3B) *p* = 0.015, with CD47 expression being higher in the earlier stage. This was true when comparing the FIGO 1 stage with FIGO 2 stage. There was no statistical significance when comparing the FIGO 1 stage with FIGO 3 stage, or the FIGO 2 stage with FIGO 3 stage.

Lymph node involvement by the process did not alter CD47 expression levels (Figure 3C). When assessing the N status at different T stages, higher expression levels were found for pT1BN0 vs. pT2N0 (*p* = 0.0096) and pT1BN1 vs. pT2N1 (*p* = 0.0085) (Figure 3D).

When we used the Allred score (Figure 4), we achieved similar results: the Allred score is higher for pT1b compared with pT2 with *p* = 0.0013 (Figure 4A); when comparing expression levels according to Allred score versus FIGO stage (Figure 4B), again, the difference was statistically significant (*p* = 0.02) when comparing FIGO 1 stage with FIGO 2 stage. There was no statistical significance when comparing the expression between FIGO 1 stage and FIGO 3 stage and FIGO 2 stage and FIGO 3 stage.

According to our results, Allred score does not depend on nodal status (Figure 4C) and it is higher for pT1b independently of nodal status (Figure 4D).

When we used the combined score (Figure 5), we achieved the following results:

It was higher for pT1b compared with pT2 (*p* = 0.0002) with no statistically significant difference between pT2A and pT2B (0.89) (Figure 5A).

When comparing CD47 expression levels by combined score versus FIGO stage (Figure 2B), again, the difference was statistically significant (*p* = 0.02), as this applies when comparing FIGO 1 stage with FIGO 2 stage. There was no statistical significance when comparing the expression of the FIGO 1 stage and FIGO 3 stage and FIGO 2 stage and FIGO 3 stage (*p* = 0.2) (Figure 5B).

According to our results, combined score does not depend on nodal status (*p* = 0.84) (Figure 5C), and it is higher for pT1b independently of the nodal status (Figure 5D).

## 4. Discussion

The tumor microenvironment (TME) is a complex ecosystem comprising various cellular and extracellular components. Cellular components include tumor cells (they influence the TME and are influenced by it); immune cells—tumor-infiltrating (lymphoid and myeloid cells that can stimulate or inhibit the antitumor immune response) and stromal cells—tumor-associated fibroblasts and endothelial cells that contribute to the structural integrity of the tumor [7,8,9,10]. Extracellular components include cytokines, hormones, the extracellular matrix, and growth factors that surround tumor cells as a vascular network [11]. The TME has a major role in the growth and development of tumors [12,13], with different cells having a strictly defined function. Endothelial cells are key in tumor development and the protection of tumor cells from the immune system—tumor angiogenesis extends beyond normal blood vessels [14], and thus provides nutritional support for tumor development. Fibroblasts promote tumor angiogenesis and the distant metastasis of tumor cells [15].

Immune cells are granulocytes, lymphocytes, and macrophages, with macrophages having a major role in immune processes in the TME [14,16].

Macrophages are the main cells of the innate immune system and perform various functions related to the development and progression of cancer; they support the extravasation of tumor cells into the circulatory system and thus ensure distant metastasis; and they can suppress antitumor immune mechanisms and responses [16]. These macrophages are defined tumor-associated macrophages (TAM), and are derived from peripheral blood monocytes from the bone marrow and differentiate into different macrophage subsets in the TME [17]. TAMs can be divided into two phenotypes: M1 and M2 macrophages [18]. M1s synthesize pro-inflammatory cytokines such as tumor necrosis factor-α (TNF-α), IL-1, IL6, IL-12, IL-23, and reactive nitrogen and intermediate oxygen compounds, and thus inhibit tumor development [19]. On the other hand, the M2 phenotype secretes cytokines such as IL-4, IL-13, IL-10, vitamin D3, and glucocorticoids, which leads to weakening of the antitumor activity and an enhancement of the ability to support angiogenesis and tissue remodeling, which is beneficial for tumor growth and invasion [20,21].

In order for M1 phenotype macrophages to perform their main activity, i.e., phagocytosis, they must recognize the tumor cell; however, tumor cells try to avoid macrophages.

Tumor cells evade immunological surveillance in three ways: loss of antigenicity; loss of immunogenicity; and the modulation of an immunosuppressive microenvironment [22].

Loss of immunogenicity can be observed even with completely preserved antigenicity and an intact neoantigen processing and presentation pathway. Typical mechanisms for reduced immunogenicity are the overexpression of negative coreceptors by T-lymphocytes and their ligands on the surface of tumor cells. One such mechanism is the expression of CD47.

The TME not only plays a key role during tumor initiation, progression, and metastases, but it also has a profound effect on the therapeutic efficacy. TME-mediated resistance to chemotherapy results from complex interactions between tumor cells and their environment [12,13].

The expression of CD47 on non-malignant cells sends a “don’t eat me” signal to phagocytes, thus ensuring immune tolerance in the human organism [23]. When CD47 is expressed on tumor cells, it enables them to evade the immune system [24,25]. For the first time, the increased expression of CD47 in malignancies was reported in ovarian carcinoma [26,27], and was later confirmed in various malignant diseases: acute myeloid leukemia (AML), non-Hodgkin’s lymphoma (NHL), breast cancer, melanoma, leiomyosarcoma, osteosarcoma, and is associated with their worsening forecast [28,29,30,31,32,33]. CD47 is known to promote the growth, invasion, and migration of cancer cells [34].

In breast carcinoma and small-cell lung cancer, CD47 expression has been reported to be associated with advanced stage at diagnosis, lymphogenous metastasis, and recurrence [24,34]. High CD47 expression has a limited correlation with reduced 5-year disease-free survival [34,35]. Using xenotransplantation models, it has been shown that anti-CD47 antibodies inhibit tumor growth and metastasis [36]. The silencing of CD47 by siRNA inhibits melanoma growth and its lung metastases [30]. The downregulation of CD47 inhibits tumor growth, cell invasion, and metastasis in non-small-cell lung cancer [24]. The overexpression of CD47 in ovarian cancer cell lines promotes cancer cell growth and motility [37].

From all that has been reported thus far, the opinion is that CD47 has a very important role in oncogenesis in many malignant diseases. To date, there have been no molecular biological studies of CD47 expression in cervical carcinoma. From our research, we can draw the following conclusions:

The results were similar regardless of which immunohistochemical method was used. The most significant correlation was observed when using the H-score (*p* = 0.00018), compared with the Allred score (*p* = 0.0013) and combined score (0.0002).

The expression of CD47 is higher for pT1b compared with pT2, and there is no statistically significant difference between pT2A and pT2B.

The expression of CD47 is higher FIGO 1 stage than FIGO 2 stage, and there is no statistical significance when comparing the expression of FIGO 1 stage and FIGO 3 stage and FIGO 2 stage and FIGO 3 stage.

The expression of CD47 does not depend on nodal status.

At first glance, these results may not be logical and diverge from those reported thus far in the literature for other neoplasms. However, our results may reflect the differential role of tumor escape mechanisms at different stages of cervical cancer evolution. One can speculate that in the early phase of cervical cancer development, tumor cell populations escape predominantly innate immunity-mediated surveillance through the up-regulation of CD47. Once this first line of immune surveillance is evaded, the cancer cell population will no longer benefit from CD47 up-regulation, but will rather need to escape T-cell-mediated eradication. Therefore, more advanced tumors down-regulate HLA class I molecules expression [38] and will no longer up-regulate CD47 expression, as demonstrated by our findings.

## 5. Conclusions

The results showed a significant difference in CD47 expression in pT1B versus pT2 patients. There was no significant difference between pT2A and pT2B. The expression of CD47 does not depend on nodal status, and it is higher for pT1b independently of the nodal status.

The most appropriate method for determining this expression is the use of the H-score.

## Figures and Tables

**Figure 1 diagnostics-13-00052-f001:**
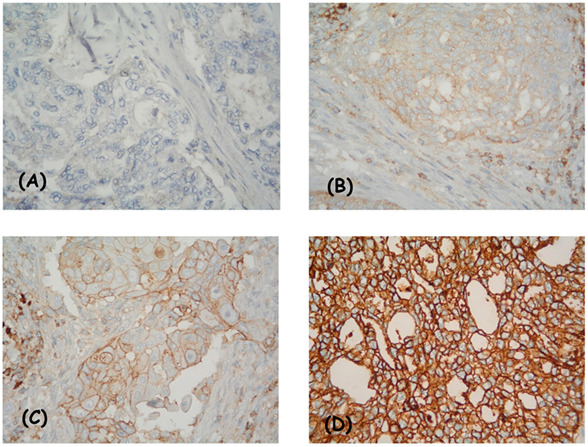
IHC expression model of CD47 in patients with cervical adenocarcinoma: membrane positivity—missing (**A**), weak (**B**), average (**C**), and strong intensity (**D**). Magnification ×400.

**Figure 2 diagnostics-13-00052-f002:**
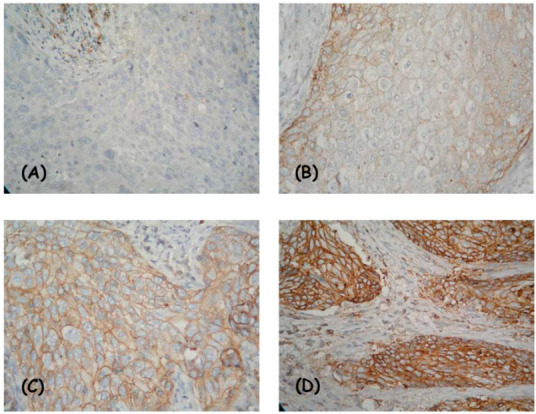
IHC expression model of CD47 in squamous cell cervical cancer: membrane positivity—missing (**A**), weak (**B**), average (**C**), and strong intensity (**D**). Magnification ×400.

**Figure 3 diagnostics-13-00052-f003:**
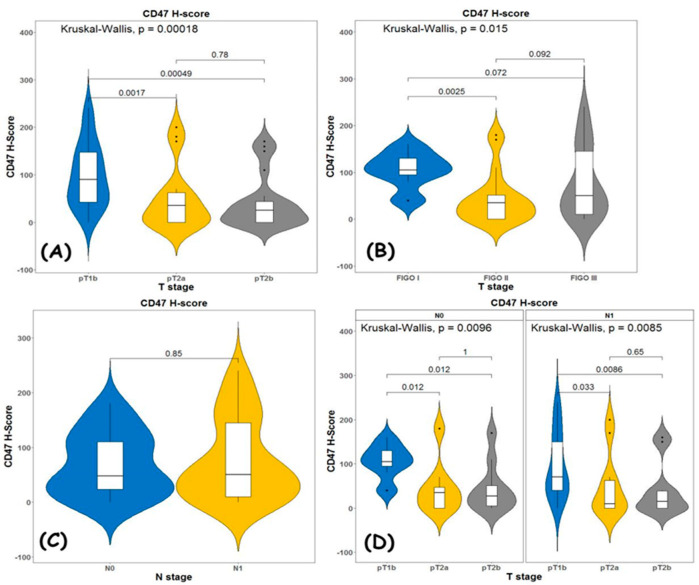
Kruskal–Wallis test for association between (**A**) TNM T stage and H-score; (**B**) FIGO stage and H-score; (**C**) N status and H-score and (**D**) T stage and H-score in node negative and node positive patients.

**Figure 4 diagnostics-13-00052-f004:**
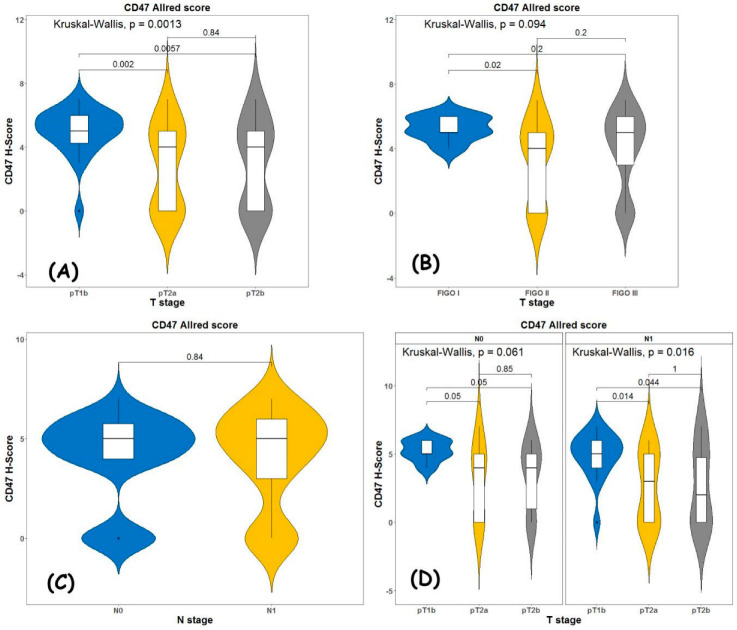
Kruskal–Wallis test for association between (**A**) TNM T stage and Allred score; (**B**) FIGO stage and Allred score; (**C**) N status and Allred score and (**D**) T stage and Allred score in node negative and node positive patients.

**Figure 5 diagnostics-13-00052-f005:**
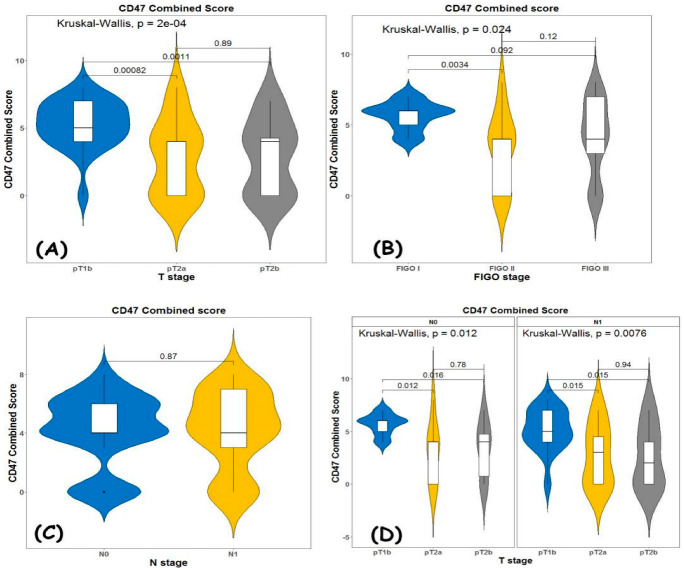
Kruskal–Wallis test for the association between (**A**) TNM T stage and combined score; (**B**) FIGO stage and combined score; (**C**) N status and combined score; and (**D**) T stage and combined score in node negative and node positive patients.

**Table 1 diagnostics-13-00052-t001:** Characteristics of patients.

Patients’ Characteristics	N	(%)
FIGO I	13	15.1
FIGO II	18	20.9
FIGO III	55	64.0
T1b *	47	54.7
T2a	19	22.1
T2b	20	23.3
N0	31	36.0
N1	55	64.0
Squamous-	78	90.7
Adenocarcinoma **	8	9.3
Total	86	100

* includes stage T1b1, T1b2, T1b3. ** this group includes also one patient with adenosquamous carcinoma.

**Table 2 diagnostics-13-00052-t002:** Allred score.

Intensity Score	Proportion Score (% Stained Cells)
0 (no staining)	0 (no cells)
1 (weak staining)	1 (<1%)
2 (moderate staining)	2 (1–10%)
3 (strong staining)	3 (11–33%)
	4 (34–66%)
	5 (67–100%)

Total score (TS) = PS + IS, TS range = 0, 2–8. TS 0 and 2 were considered negative. Scores of 3–8 were considered positive.

**Table 3 diagnostics-13-00052-t003:** Distribution of patients according to FIGO stage and the final immunohistochemical scores.

Histologic Results Stage	H-Score			Allred Score		Combined Score		
	Negative	Low Expression	High Expression	Negative	Positive	Negative	Weak Expression	Overexpression
FIGO Stage I	0	2	11	0	13	0	11	2
FIGO Stage II	6	9	3	6	12	6	10	2
FOGO Stage III	12	23	20	12	43	12	28	15

## Data Availability

The authors declare that all related data are available from the corresponding author upon reasonable request.

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
