# Peer review of "Prognostic Utility of CD47 in Cancer of the Uterine Cervix and the Sensitivity of Immunohistochemical Scores"

_diagnostics, 2022, doi:10.3390/diagnostics13010052_

Round 1

Reviewer 1 Report

the study is approached in a correct and detailed way. Although the results are not those expected, the work is still well done.

Author Response

Thank you very much for reviewing and approving the article. I greatly appreciate you taking the time to review it!

Reviewer 2 Report

Very good idea to check the potential of CD 47 as a prognostic factor in cancer.

I think it is worth to check its potential in the bigger group of cases.

Author Response

Thank you very much for reviewing and approving the article. I greatly appreciate you taking the time to review it!

Now we start the new project and one of it's aims is to  exam the CD47 expression in larger group. 

Reviewer 3 Report

Well presented and documented manuscript of scientific and clinical interest

Author Response

(The authors gave the same response as above.)
